

# G6-1.5K-SAI: a new Geoengineering Model Intercomparison Project (GeoMIP) experiment integrating recent advances in solar radiation modification studies

Daniele Visioni[1], Alan Robock[2], Jim Haywood[3,4], Matthew Henry[4], Simone Tilmes[5], Douglas G. MacMartin[6], Ben Kravitz[7,8], Sarah Doherty[9], John Moore[10], Chris Lennard[11], Shingo Watanabe[12], Helene Muri[13], Ulrike Niemeier[14], Olivier Boucher[15], Abu Syed[16], and Temitope S. Egbebiyi[17]

[1]Department of Earth and Atmospheric Sciences, Cornell University, Ithaca, NY, USA
[2]Department of Environmental Sciences, Rutgers University, New Brunswick, NJ, USA
[3]Met Office Hadley Centre, Exeter, UK
[4]College of Engineering, Mathematics and Physical Sciences, University of Exeter, Exeter, UK
[5]National Center for Atmospheric Research, Boulder, CO, USA
[6]Sibley School of Mechanical and Aerospace Engineering, Cornell University, Ithaca, NY, USA
[7]Department of Earth and Atmospheric Science, Indiana University, Bloomington, IN, USA
[8]Atmospheric Sciences and Global Change Division, Pacific Northwest National Laboratory, Richland, WA, USA
[9]CICOES (Cooperative Institute for Climate, Ocean and Ecosystem Studies), University of Washington, Seattle, WA
[10]Arctic Centre, University of Lapland, Rovaniemi, 96101, Finland
[11]Climate System Analysis Group, University of Cape Town, Cape Town, South Africa
[12]Japan Agency for Marine-Earth Science and Technology, Natsushimacho, Yokosuka, Kanagawa, Japan
[13]Department of Energy and Process Engineering, Industrial Ecology Programme, Norwegian University of Science and Technology, Trondheim, Norway
[14]Max Planck Institute for Meteorology, Hamburg, Germany
[15]Institut Pierre-Simon Laplace, Sorbonne Université/CNRS, Paris, France
[16]Centre for Rediscovered and Redefined Natural Resources Research and Education (C4RE), Dhaka, Bangladesh
[17]Dept. of Environmental and Geographical Science,University of Cape Town, Cape Town, South Africa

**Correspondence:** Daniele Visioni (dv224@cornell.edu)

**Abstract.** The Geoengineering Model Intercomparison Project (GeoMIP) has proposed multiple model experiments during the phases 5 and 6 of the Climate Model Intercomparison Project (CMIP), with the latest set of model experiment proposed in 2015. With phase 7 of CMIP in preparation, and with multiple efforts ongoing to better explore the potential space of outcomes for different Solar Radiation Modification (SRM) both in terms of deployment strategies and scenarios and in terms of potential

impacts, the GeoMIP community has identified the need to propose and conduct a new experiment that could serve as a bridge between past iterations and future CMIP7 experiments. Here we report the details of such a proposed experiment, named G6-1.5K-SAI, to be conducted with the current generation of scenarios and models from CMIP6, and clarify the reasoning behind many of the new choices introduced. Namely, compared to the CMIP6 GeoMIP scenario G6sulfur, here we decided on: 1) an intermediate emission scenario as baseline (the Shared Socioeconomic Pathway 2-4.5); 2) a start date set in the

future that includes both considerations around the likelihood of exceeding 1.5ºC above preindustrial and some considerations around a likely start date for an SRM implementation; 3) a deployment strategy for Stratospheric Aerosol Injection that does not inject in the tropical pipe in order to obtain a more latitudinally uniform aerosol distribution. We also offer more details over the



preferred experiment length and number of ensemble members, and include potential options for second-tier experiments some modeling groups might want to run. The specifics of the proposed experiment will further allow for a more direct comparison

between results obtained with CMIP6 models and those obtained with future scenarios for CMIP7.

## 1 Introduction

The Geoengineering Model Intercomparison Project (GeoMIP) was set up in 2011 (Kravitz et al., 2011) as a way to standardize climate model experiments of Solar Radiation Modification (SRM), a form of climate intervention (or geoengineering) that aims to reduce surface temperatures by means of preventing a portion of the incoming solar radiation from reaching the surface.

This could be achieved by a variety of proposed techniques, of which many have been explored through GeoMIP (Visioni et al., 2023b). Standardized experiments help diagnose the potential sources of differences between models' responses to SRM, and are therefore a necessary step to better identify areas of agreement and disagreement, and areas where models can be improve. This has been done in general for experiments related to climate change since the Coupled Model Intercomparison Project (CMIP) (CMIP; Meehl et al., 2005). For CMIP, this standardization takes the form of prescribing the same concentrations, or

emissions, of greenhouse gases and other climate-altering factors (such as land use changes and aerosols), both for historical conditions and for future ones through the Scenario Model Intercomparison Project, ScenarioMIP (Meinshausen et al., 2020). For SRM, the underlying emission scenario is one of the things that needs to be prescribed, but on top of that the specifics of the climate intervention need to be specified as well. This implies deciding on the way in which radiation is altered (from the simplest reduction in the top-of-atmosphere solar constant, to the injection of $SO_2$ in the stratosphere, to a prescribed increase

in ice crystals fall velocities to reduce cirrus cloud optical depth), the SRM strategy, and when to start the simulated intervention and how much radiation to alter, and eventually when to stop, the SRM scenario.

In Visioni et al. (2023b), we took stock of the previous decade and more of GeoMIP experiments, reviewing both official "Tier 1" experiments that were part of phases 5 and 6 of CMIP and also parallel experiments produced by the GeoMIP

community in order to better identify some sources of uncertainty for SRM and to explore other potential scenarios than those prescribed in CMIP5 and CMIP6. The discussion continued during the annual GeoMIP meeting held in Exeter during the summer of 2023 (with a summary of the meeting presented in Visioni et al. (2023c)), and mostly focused on potential future experiments that will need to be run as part of the next, seventh iteration of CMIP (CMIP7). During such discussions, the community has identified some pressing needs that have to be considered when thinking about future experiments, and that

will constitute the target for the experiment we are proposing here:

1. Having traceable, simple experiments that can remain consistent across different iterations, in order to understand changes and improvements in Earth System Models (ESMs) and how models' differences evolve over time and as ESMs become more complex. For instance, the experiment G1 is a very simple experiment that involves reducing the solar constant in order to prevent temperatures from changing under a 4xCO$_2$ scenario, and has been successfully per-



formed across various generations of ESMs (Kravitz et al., 2021), and using models with very different resolution and characteristics (Virgin and Fletcher, 2022).

2. Considering novel experiments that build on past gathered knowledge (gained through GeoMIP experiments, or through other, related, experiments) to improve, clarify and expand the potential space of SRM scenarios. For instance, experiments G3 and G4 in CMIP5 for Stratospheric Aerosol Injection (SAI) considered equatorial injections of $SO_2$ in order to

more closely mimic the 1991 Mt. Pinatubo volcanic eruption (Kravitz et al., 2011; Berdahl et al., 2014), while G6sulfur in CMIP6 considered injections uniformly between 10ºN and 10ºS (Kravitz et al., 2015; Visioni et al., 2021b). Following research has higlighted that extra-tropical injections have different impacts than tropical injections, and specifically avoid some of the identified negative climate responses to tropical injections (Kravitz et al., 2019; Visioni et al., 2021a). Similarly, for Marine Cloud Brightening (MCB) there have been multiple lines of research pursued after the G4sea-salt

and G4cdnc experiments (Alterskjaer et al., 2013; Ahlm et al., 2017) that moved away from broad injections over entire latitudinal bands and towards the injection of sea salt over specific, susceptible areas (Haywood et al., 2023b).

3. Having experiments that are up-to-date in terms of 'policy relevance', in the sense of considering SRM under future scenarios that are of interest to the scientific community, informative and plausible (MacMartin et al., 2022), which means keeping up-to-date with current emission or concentration scenarios as considered by ScenarioMIP (Meinshausen

et al., 2020, 2023). This is also relevant when discussing efforts aimed at considering local impacts of interest for different communities, for instance in terms of ecological impacts (Zarnetske et al., 2021) or regional climatic changes (Kuswanto et al., 2022).

Different parts of the communities might give more or less weight to such needs, especially to the tension between process understanding and policy relevance, but a balance needs to be found if future devised experiments are to remain, in the broadest

sense, useful. During the meeting, a proposal was put forward for an experiment to be run that is capable of addressing some of these needs and whose protocol we set out to describe in this manuscript. Namely, this experiment is to be conducted with the current generation of Earth System Models and scenarios that participated in CMIP6, but builds upon novel findings around SRM to constitute an intermediate experiment capable of informing upcoming decisions around CMIP7 experiments. As we will discuss, such an experiment also ensures a high degree of comparability with future CMIP7 scenarios.

**2   Reasons behind a new experiment and its timing**

CMIP6 GeoMIP experiments were originally proposed in 2015 (Kravitz et al., 2015). Eight years later, it is useful to reconsider and potentially update the scenario choices that have been performed at the time. In those years, there have been multiple discussions in the climate science community with regards to the plausibility of some specific future climate scenarios such as SSP5-8.5, on which G6sulfur is based (Burgess et al., 2020). Furthermore, G6sulfur has a set start date in 2020, which has

passed, and so is clearly unrealistic. Finally, those scenario choices are contemporary with the decisions taken in the Paris Agreement and precede the Intergovernmental Panel on Climate Change (IPCC) special report on 1.5ºC (Masson-Delmotte





et al., 2018). It is useful to reiterate that these observations alone do not discount or invalidate research done, or in progress, using such scenario. Interesting and useful research can be done, and is still being done, even with older scenarios like G4 (Chen et al., 2020; Kuswanto et al., 2022). Nonetheless, a scenario that might be considered more "realistic" in terms of those

factors (scenario choice, starting year, SRM target) might be of use to many. Members of the GeoMIP community have also contributed to international reports, such as the IPCC Sixth Assessment Report (AR6, Chen et al. (2021)) and the World Meteorological Organization (WMO) 2022 Ozone Assessment (Haywood et al., 2023a), with GeoMIP results, leading to numerous insights over what constitutes a useful scenario in the case of those reports.

Since 2015, there have also been multiple advances in terms of our understanding of the potential impacts of different forms of SRM. For SAI, there have been multiple investigations highlighting the importance of injection location (Tilmes et al., 2017; Kravitz et al., 2019; Visioni et al., 2021a) and cooling target (Irvine et al., 2019; Lee et al., 2021) in the determination of the impacts. For MCB, there have been multiple advances in terms of where to brighten clouds, and which size would be necessary for the injected particles (Wood, 2021; Haywood et al., 2023b). Such new knowledge should be integrated in the future set of

GeoMIP experiments.

Both of these points could lead to the conclusion that such decisions should be deferred to the next set of GeoMIP experiments for CMIP7, however, there is one main reason why we are proposing this intermediate experiment now. It is extremely likely, based on timelines provided by CMIP in the summer of 2023, that the next set of scenario forcings from ScenarioMIP

will not be available before early 2026, meaning that the first set of CMIP7 GeoMIP results might come as late as 2028 given priorities from the modeling centers. This would mean a gap of almost 10 years between when GeoMIP CMIP6 simulations were released and the CMIP7 ones, which, based on the numerous calls for more research into SRM from national and international organizations (National Academies of Sciences Engineering and Medicine, 2021), would be a large gap. An intermediate experiment to be performed over late 2023 and 2024 with multiple models could fill this gap, and allow for more informed

decisions moving towards CMIP7.

## 3   Required decisions towards a new experiment

In this section we aim to list all aspects that need to be decided when constructing a new GeoMIP experiment. There are multiple discussions of scenario-building and of the relevance of scenarios, in the context of climate change and of SRM, available in the literature (Parson, 2008; Bellamy et al., 2013; O'Neill et al., 2016; MacMartin et al., 2022; Diamond et al.,

2022), and past experimental protocols from GeoMIP also include the explicit mention of some of these decisions (Kravitz et al., 2011, 2015). In the following list however we give an overview of a more conclusive list of all the decisions that need to be made, particularly in the context of multi-model experiments, with a list of potential different choices, while in the next section we will explain why we made the particular choices for this specific experiment.



1. **Metrics**. Deciding on a metric means selecting a target quantity on which to base decisions around the simulated deployment of SRM. Not all SRM simulations necessarily have a target metric, for instance the experiment G4 injected a fixed amount of $SO_2$ for a number of years. However, most simulations do: the various generations of GeoMIP experiments have either used global mean surface air temperature (GMSAT) or top-of-atmosphere radiative forcing (TOARF) as the metric against which SRM is assessed. Originally, G6sulfur aimed at reducing radiative forcing from SSP5-8.5 to a SSP2-4.5 target, but practicalities in simulations meant that this was soon modified from radiative forcing to a temperature target which was defined as achieving the target temperature within a decadal mean of $\pm$ 0.2 K. Therefore, a successful G6sulfur simulation was one in which GMSAT was the same as that in SSP2-4.5 target within those limits, and other global or regional quantities could be compared against that. GMSAT and TOARF are easy metrics to compute, and either directly related to the idea itself of SRM (for TOARF) or to climatic targets as those defined in the Paris Agreement (for GMSAT), with a robust scientific basis beyond them (Knutti et al., 2016), but they are by no means the only possible metrics. Global precipitation-based metrics have been proposed (Lee et al., 2021), and so have metrics that go behind global mean values, for instance targeting also inter-hemispheric and equator-to-pole temperature gradients (Kravitz et al., 2016). It would also be legitimate to choose metrics that are regionally based (for instance, precipitation changes over a specific region), more directly based on agricultural or economic metrics (Clark et al., 2023), or metrics that integrate multiple quantities in a more comprehensive way (for instance, Song et al. (2022) discussed the concept of a surface equivalent potential temperature metric for global warming). More studies focusing on those other metrics could be useful to inform future decisions in regards to GeoMIP experiments. Lastly, it is useful to note that a similar framework as G6sulfur could be harder to achieve in CMIP7 if models move to emission-driven scenarios (Meinshausen et al., 2023) in which $CO_2$ concentrations (and therefore forcing) are harder to compare across underlying scenarios, if the carbon cycle is allowed to change due to warming or cooling. Therefore, for future simulations, a target that is not related to other scenarios (so, for instance, 1.5ºC or, 2ºC above pre-industrial (PI) GMSAT) would be much easier to implement.

2. **Underlying emission scenario**. Choosing an underlying emission scenario implies choosing the amount of intervention, connected also to the chosen target. For instance, G6sulfur used SSP5-8.5, with the main aim of obtaining a good signal to noise ratio in order to achieve better process understanding. However, especially when choosing a non-idealized emission scenario (that is, a specific SSP as opposed to a 1%$CO_2$ increase) means sometimes having to contextualize SRM in that scenario. For instance, the SSP5-8.5 emission pathway has been criticized in the literature for being unrealistic in many respects (Burgess et al., 2020). It would also not be a preferable scenario under which one should imagine a climate intervention strategy, due to the lack of emission abatement and risks of termination shocks (Zarnetske et al., 2021). Finally, an emission scenario similar to SSP5-8.5 is unlikely to be repeated for CMIP7 (Meinshausen et al., 2023). Therefore, selecting a new underlying emission scenario that will be repeated (at least in a similar form) in CMIP7 would be preferable.



3. **SRM Start date**. The date when SRM is started in a GeoMIP experiment should not be interpreted as a prediction or a recommendation of when SRM would start. As noted before, G6sulfur considered a date of 2020 for its start, which has now passed. Nonetheless, at least two of the models that participated in G6sulfur did not start injections until 2030 or 2040 (Visioni et al., 2021b), given that GMSAT between SSP2-4.5 and SSP5-8.5 were indistinguishable until later decades. A new starting date for future experiments that moves away from specific years and that takes into account information such as the likelihood of crossing 1.5ºC or 2ºC above PI, while also taking into account the feasibility of a given implementation being scaled up as specified in the simulation, would remove some ambiguity.

4. **SRM strategy**. Since 2015, many studies of SAI have shown that strategies that move away from equatorial injections, as was used for G6sulfur, might be preferable. Recently, Henry et al. (2023) have compared two models using a controller (Kravitz et al., 2016) to manage four injection locations (30ºN, 15ºN, 15ºS, 30ºS). However, this would be hard to achieve for models which have not implemented a feedback controller. Furthermore, Zhang et al. (2023) pointed out that a 4-location controller managed injection strategy might be similar to the outcomes of a simpler, 30º N and 30º S symmetrical strategy. Questions remain as to whether such an experiment (in terms of temperature targets) should also be performed through GeoMIP for MCB as the residual climate response for currently proposed MCB simulations is likely to be very much less homogeneous than for SAI (e.g., Haywood et al. (2023b)). Furthermore, there are many more degrees of freedom in how MCB could be implemented, and very different cloud fractions and cloud albedo susceptibilities across different models, showing that much work needs to be done to figure out how to specify an MCB scenario that can be similarly implemented across models. At this point, there is probably little value in running another solar dimming-like experiment as there are specific dynamical feedbacks, impacts on stratospheric ozone, and differences in response of crops and natural vegetation to direct and diffuse radiation that appear important for stratospheric aerosols (e.g. Jones et al. (2021); Visioni et al. (2021a)), and this is clearly not a good proxy for MCB, which would have very regionally-focused forcings. As for the start date, for any specific SRM strategy there are questions around their feasibility in a technological or geo-political sense in terms of injection location, targets, injection altitude and scalability.

5. **Length of experiment** The G6sulfur simulations were run out to 2100, for the main reason that that was the end date for most CMIP6 forcing datasets. Decisions around simulation length should account for the crucial question of what the actual point of the experiment is. If the purpose is detecting the time of emergence of the SRM signal (which could be of the order of a decade, globally, or more regionally, depending on the magnitude of the forcing, Keys et al. (2022)), prioritizing ensemble size over length would be preferable. If it is to understand the long-term Earth system model response to SRM, then one should prioritize longer runs (decades to century timescales). If it is to understand near-term climate change for climate policy decision making (a few decades), a mix of the two priorities may be appropriate. A good way to frame the question should be, "If a modeling center only has 100 years of simulation time available, how should they preferentially be used?" For example, one could prefer 3 ensemble members for 35 years (as suggested in MacMartin et al. (2022)), rather than one ensemble member for 100 years. Some precedent for running longer simulations exists from the CMIP6 simulations; for example, many climate modeling centers ran the overshoot scenario SSP5-3.4OS with





multiple ensemble members until 2100, but a single member through to 2300. The results show significant differences from the simplified representation of overshoot expressed in many studies (e.g., Geden and Löschel (2017)) and would suggest that SRM may need to be maintained for long periods of time in order to achieve temperature targets such as 1.5°C or 2°C above PI (Baur et al., 2023).

All of these necessary choices have been summarized in Figure 1. In the figure we have included multiple potential tiers of experiments (intending "Tier 2" as lower-priorities ones) to be as generic as possible, to suggest a flexibility in the framework to allow a subset of groups to run variants that can leverage specific tools or capabilities in individual models.

## 3.1     Reflections on community engagement on how to make scenario-related decisions in GeoMIP

The large attendance at the 13th GeoMIP meeting highlighted the extent to which the core group of climate modelers who orig-
inally devised GeoMIP has expanded to many more interested users and parties, including researchers interested in ecological and societal impacts, researchers from the Global South concerned with specific regional impacts, and researchers interested in climate emulators. Hence, finding common ground for a scenario with which everyone agrees is difficult. For example, designing an emulator would require a multitude of simulations to provide training data; such an approach has been taken in emulating explosive volcanic eruptions (Aubry et al., 2020). On the other hand, understanding regional impacts such as
precipitation changes over South Asia or Africa requires a more policy-relevant scenario. Importantly, a scenario that a part of the community might find interesting might not be a scenario that climate modelers themselves find desirable to prioritize. All these reflections have been expanded upon in the related meeting report (Visioni et al., 2023c).

## 4     Experiment proposal for G6-1.5K-SAI

What follows is the initial proposal for a new GeoMIP experiment, hereby named "G6-1.5K-SAI", selecting choices for all the
open questions in Figure 1. Close to each "decision" (in bold) there is an explanation for why that decision might be "optimal" from the point of view of GeoMIP, and an exploration of potential other choices and why we did not take them. A summary figure is provided in Fig. 3 below.

1.  **Target metric: GMSAT** The Paris Agreement is defined in terms of breaching or not a GMSAT metric; many parts of the latest IPCC reports (Masson-Delmotte et al., 2018; Chen et al., 2021) discuss changes in regional climate and in impacts
with respect to global mean temperature, and many of those scale linearly with GMSAT increases (Knutti et al., 2016; Seneviratne et al., 2016). Other proposed metrics, such as global mean precipitation (GMP), might be easily derived from GMSAT. For instance, in the ARISE simulations (Richter et al., 2022) the target for SAI was 1.5°C, which corresponded to the 2020-2039 average. The corresponding GMP for that intervention during the 2050-2069 period (2.94 mm/day) was only slightly below the value for the 2020-2039 average (2.95 mm/day), while the corresponding value for the same future
period under the underlying emission scenario was 3.01 mm/day. In general, also for larger cooling, the warming-driven precipitation increase is larger than the SRM-specific precipitation decrease (Visioni et al., 2023a) in a global sense, and



while the two cannot be controlled simultaneously (Lee et al., 2021), there is always a relationship between global mean temperature changes and global mean precipitation changes (the hydrological sensitivity, Pendergrass (2020)), meaning that, based on simulations that target GMSAT, the equivalent results for hypotetical simulations that target GMP can easily be found by scaling the GMSAT results. The same error margins as G6sulfur of $\pm 0.2$K in the decadal mean should be considered.

2. **Underlying emission scenario: SSP2-4.5** Of all the current CMIP6 scenarios, SSP2-4.5 is the one understood to be closest to current emission pledges, especially in the medium term (see discussion MacMartin et al. (2022) and Plummer et al. (2021)). Therefore, it might be considered as one of the most "policy relevant" scenario in which one would be interested to understand SRM impacts. It is also worth considering that in the pre-2050 timeframe all SSP emission scenarios look globally very similar as a consequence, and so does the resulting GMSAT from most climate models (Tebaldi et al., 2021). A scenario similar to SSP2-4.5 is also expected to be central to CMIP7 (Meinshausen et al., 2023). During the 13th GeoMIP meeting, the question of the potential use of an overshoot scenario in GeoMIP simulations was also discussed (see Visioni et al. (2023c)). The current overshoot scenario that has been performed under CMIP6 - SSP5-3.4OS - is a possibility, as described in Tilmes et al. (2020). Currently, 4 out of 6 of the models that participated in G6 have also simulated SSP5-3.4OS; of these four, only a fraction of the variables provided for SSP2-4.5 are available (from 40% for CESM2-WACCM and UKESM1 to 10% for IPSL). Therefore, it might be challenging for modeling centers that need to re-run the simulations, and therefore the climate impacts community might have problems finding the data they need. Finally, if short-term simulations are considered, SSP5-3.4OS does not look that much different from SSP5-8.5 in Tilmes et al. (2020): in 2050, the SAI injection rate needed to stay at 1.5ºC is 12 Tg $SO_2$/yr for both scenarios.

3. **1.5ºC above pre-industrial (using definition 3 below)** 1.5ºC is a meaningful target for the Paris Agreement and has been widely used in the latest simulations (i.e., Tilmes et al. (2020); Richter et al. (2022); MacMartin et al. (2022)). It also allows for different, lower priority tiers with higher (2.0ºC) or lower (1.0ºC) temperature targets. There are many ways in which one could define "above pre-industrial (PI)" in an operative way. Here we outline three possibilities: 1) use the models' PIcontrol values (which can vary), with consequences for how inter-model comparisons would be conducted since some models will reach 1.5ºC much faster than others; 2) use an externally measured value for PI to have an external and common base for all models, with the similar consequence as (1) that different models will still reach 1.5ºC at different dates; 3) use the 2020-2039 average as the definition of 1.5ºC as described in MacMartin et al. (2022), so that, given the same starting date, all models can start "ramping up" with the SRM amount independently of how fast they were in the historical period at warming. As noted by Henry et al. (2023), the choice of both 2035 and of defining 1.5ºC compared to the model PI period may mean relatively rapid deployment of SAI in models that have already exceeded the 1.5ºC target. If the start date were also changed in each model dependent on when that model reached 1.5 ºC, that may result in implausible start-dates, as well as making intermodel comparisons more difficult. Some of these differences are evident in Fig. 4, as models' PI temperatures can vary by over 1.5 ºC. On the other hand, global models' spread in 2020-2039 GMSAT is much smaller (1 K). Therefore, we conclude that definition 3 is the best



basis upon which to define the starting date across different models - even if it might not be necessarily ideal when considering experiments with one single model (as in Tilmes et al. (2020), which used option 1). Different temperature targets can still be considered and included as secondary tiers for interested modeling centers. For some, given the fact that observed GMSAT may exceed the 1.5ºC target in the next decade and considering the significant development times for any practical deployment of SAI, a GMSAT target of 2ºC might be considered more pragmatic. This may address the request put forward during the latest meeting to have multiple scenarios to compare when doing SAI assessments.

4. **Start date of simulation and SRM implementation: 2035** This start date is easier to justify if the 2020-2039-based definition of 1.5ºC (option 3 above) is used; a time-frame of over 10 years before a deployment could also be a reasonable guess for when a scaled-up deployment may conceivably start. Combined, the two choices allow for a slower "ramping up" of injections as opposed to lower temperature targets (Visioni et al., 2023a) which require much more cooling at the beginning. Later dates could be considered, but then how fast the cooling should be achieved would need to be properly defined as well: MacMartin et al. (2022) selected a 10 year period, but this is arbitrary (and for climate velocities in relationship to ecosystems resilience, it might be too high; Trisos et al. (2018)).

5. **End date of simulation: 2085 (50 years after beginning)** As described in the previous section, the appropriate end date strongly depends on research priorities. If the community is more interested in signal emergence, and the modeling groups have limited computational capabilities, then 50 years should be prioritized to run three shorter ensemble members of 50 years (rather than one for 150 years). If modeling teams have more computer time, one ensemble member could be extended to 2100 to explore longer term impacts like sea level rise and tipping points. At the end of the decided timeframe, some models might be interested to look at the effects of a "phase down" (MacMartin et al., 2022), or a termination, as done in the experiments G2 (Jones and Haywood, 2012) and G4 (Trisos et al., 2018). This should not be included in the Tier 1 experiment, that should end in 2085, but should be treated as a "Tier 2" branch run with different conditions from the main one, and a different name for the experiment.

6. **Forcing strategy for SRM method: SAI at 30ºN and 30ºS, symmetrical at 21 km** As of now, not many models are able to include a controller for SAI capable of managing multiple injection locations and targets; therefore, a symmetric injection strategy at 30ºN and 30ºS (one longitude, one vertical layer) seems the most feasible to avoid problems with overcooling the tropics while doing as reasonable of a job at many metrics as more complex injection strategies (see Zhang et al. (2023) for CESM2 results and Fig. 4 for a comparison with UKESM results). Injection should be of $SO_2$, with an option for prescribing optical depth. As shown for G6sulfur results (Visioni et al., 2021b), there is functionally no difference if the injection amounts are changed every one or every ten years in order to achieve the desired temperature targets in the models, but for consistency with more recent simulations, a yearly update to the injection rates should be considered when possible. The choice of altitude, similar to other recent experiments (Richter et al., 2022) but narrower than G6sulfur (that was between 19 and 21 km) offers a good compromise between lifetime (Lee et al., 2023b) and technical constraints around deployment (Smith et al., 2022). For this experiment, we have decided to not include an MCB option: currently, there is ongoing research towards better defining the potential areas where to apply the local





forcing and how to control for different targets, as it has been done with SAI previously, and the community is working towards a set of experiments that might help clarify the path forward for the next GeoMIP iterations.

## 5   Data requests for G6-1.5

Multiple groups at the latest GeoMIP meetings highlighted the need for specific data to be uploaded to be able to understand some impacts. In this section we give a brief overview of what variables in particular should be provided by the modeling
centers in order to conduct some of the analyses of interest to the community.

- *Ocean and cryosphere.* Changes in 3D ocean currents, heat content and tropospheric wind fields are extremely important when considering change in regional sea levels, hurricane potential and teleconnection patterns. Similarly, given the polar amplification underway, change in snow and sea ice cover, surface runoff, soil temperatures and measures of biological activity are also valuable to understand the behavior of potential feedbacks in the context of SRM, such as those related
to carbon release from permafrost thawing (Chen et al., 2020; Lee et al., 2023a).

- *Compound indices for health, well-being and urban planning.* Daily minimum and maximum surface air temperature and precipitation, and also possibly wind speeds and humidity can be used to construct compound indices, and provide valuable inputs to human health impact models (Song et al., 2022), and be valuable in evaluating potential urban planning scenarios dealing with, for example, flood risk. Such daily data is also necessary to build indices such as the Expert Team
of Climate Change Detection Indices (ETCDDI) for climatic extreme analysis (Tye et al., 2022; Tan et al., 2023; Patel et al., 2023) and to inform hydrological models such as the Soil and Water Assessment Tool (SWAT, Tan et al. (2023)).

- *Agricultural and ecological modeling.* To better understand SRM impacts on crops and ecological system, daily (and sometimes sub-daily) data related to changes in solar radiation (such as direct and diffuse changes) can also be of relevance, together with temperature and precipitation (Zarnetske et al., 2021). Other variables might include those
necessary to calculate sulfate deposition rates for SAI (Visioni et al., 2020), as not every model for G6sulfur uploaded them.

## 6   Conclusions - the road towards CMIP7

Here we have described a new GeoMIP experiment to be run with the current Earth System Model generation (i.e. with models that are participating in CMIP6). This new experiment proposes some novel advances in the experimental design compared to
the last iteration of GeoMIP experiments such as G6sulfur (Kravitz et al., 2015), in particular related to start date, injection strategy for $SO_2$ and considerations of recent policy-relevant targets such as those from the Paris Agreement. Furthermore, we have clearly outlined all the necessary choices that need to be made when considering an SRM modeling experiment, and openly explained each decision in relation to the scenario selected, in order to facilitate future discussions about scenarios in GeoMIP as we move towards deciding experiments for CMIP7.



The scenario choice we operated in terms of chosen target described above offers a way to maintain more consistency between CMIP6 and CMIP7 model experiments, given the direction of basing CMIP7 models on emission-driven, rather than concentration-driven, scenarios. Comparing across models' generations is a very useful exercise to understand sources of uncertainty and model disagreement, which is what made a simple experiment like G1 so successful (Kravitz et al., 2021). The current G6sulfur experiment might be harder to compare against any CMIP7 experiment, given its reliance on two SSP sce-

narios, one of which most likely will not be repeated (SSP5-8.5), while the new experiment we proposed might more easily be reproduced in CMIP7 given its middle-of-the-road scenario selected (SSP2-4.5) and temperature target independent of scenario choices.

As remarked in Visioni et al. (2023b), GeoMIP experiments do not need to encompass all potential SRM applications, and therefore we are not claiming our scenario choices indicate the only, or optimal, scenario under which SRM should be considered or studied: the main focus of GeoMIP remains to offer a robust framework for model intercomparison through standardized experiments, which means they need to remain somewhat simple compared to the complexities of any given "realistic" SRM application in the real world, in order to understand the underlying processes determining climatic impacts. More complex injection strategies than the one we proposed here, or less-then-ideal scenarios with one or multiple actors

are still an important area of research, and G6-1.5 should be considered as a useful common benchmark against which other scenarios can be tested, for instance, by a single model.

*Code and data availability.*  Data for Figure 4 is available at https://zenodo.org/record/8430485. Data for Figure 2 is available from the Earth System Grid https://esgf-node.llnl.gov/search/cmip6/. No original data as been produced for this manuscript.

*Author contributions.*  DV wrote the text and produced the figures, with contribution and editing from all other authors both during the

GeoMIP summer meeting and throughout the process. MH produced Figure 4 and assisted with data upload and analyses.

*Competing interests.*  The authors declare no competing interests.

*Acknowledgements.*  DV is supported by Cornell Atkinson Center for Sustainability. AR is supported by NSF grant AGS-2017113 and a gift from the SilverLining Safe Climate Research Initiative. MH is funded by the Natural Environment Research Council Exeter-NCAR (EX-TEND) collaborative development grant (NE/W003880/1) and by SilverLining through the Safe Climate Research Initiative. SW is supported

by the Japan Society for the Promotion of Science (grant no. JP2103668). Support for BK was provided in part by the National Science Foundation through agreement SES-1754740, NOAA's Climate Program Office, Earth's Radiation Budget (ERB) (Grant NA22OAR4310479), and the Indiana University Environmental Resilience Institute. The Pacific Northwest National Laboratory is operated for the US Department of



Energy by Battelle Memorial Institute under contract DE-AC05-76RL01830. DGM is partially supported by the National Science Foundation through agreement CBET-2038246



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



Experiment

Target metric →

Underlying emission scenario

Target

Tier 2 Target

Forcing strategy for SRM method

Start date

End date

Time (yr) →

## Decisions needed

| | | | |
|---|---|---|---|
| **Target metric** | Radiative Forcing, Global mean temperature (GMSAT), Global mean precipitation (GMP), Other impacts | **Target(s)** | "Policy relevant" threshold (1.5ºC,2ºC) Dependent on historical period (i.e. maintain GMP at 2020 levels) "Moving target" (i.e. halve the warming) |
| **Forcing strategy for SRM method** | Solar dimming SAI injection locations (latitude, altitude) SAI material ($SO_2$, $H_2SO_4$, prescribed aerosols, other materials) MCB area and method (sea salt injections or increasing cloud droplets number) Use of controller for multiple targets | **Emission scenario** | SSP2-4.5 ("Current policy") SSP5-3.4OS ("Overshoot") $1\%CO_2$ or $2xCO_2$ |
| **Start date** | Fixed date (i.e. 2035) Dependent on target (i.e. when 1.5ºC is reached) | **End date** | End of underlying scenario (2100) Maximizing ensemble members (for signal emergence) Studying long term consequences (i.e. sea level rises) |

**Figure 1.** A summary of necessary decisions for the new proposed experiments. The black line represents the underlying emission scenario (e.g., SSP2-4.5); the blue lines represent the potential targets (which depend on the chosen target metric, and do not have to be constant). The red lines represent the forcing that needs to be applied, based on the underlying emission scenario and the targets. On the right, all key decisions are listed (red boxes), followed by more concrete examples as provided in the text as well.







**Figure 2.** Global mean surface air temperature (GMSAT) in models participating in the sixth phase of GeoMIP for the historical (1850-2014) and SSP2-4.5 (2015-2100) period, showing annual means (thin lines) and 20-years running means (thick lines). b) GMSAT averages for periods relevant to the question of start and end dates for SRM experiments. PI is defined as the average, for each model, over their entire simulated PIcontrol simulations. c) Time periods in which each model's SSP2-4.5 simulation reaches PI+1.5 (considering a 20-year running average). The year 2035 (the proposed start date for PI+1.5 not considering the model PI) is indicated with a vertical dashed line. For this figure, only the first ensemble member for each model has been used for consistency.



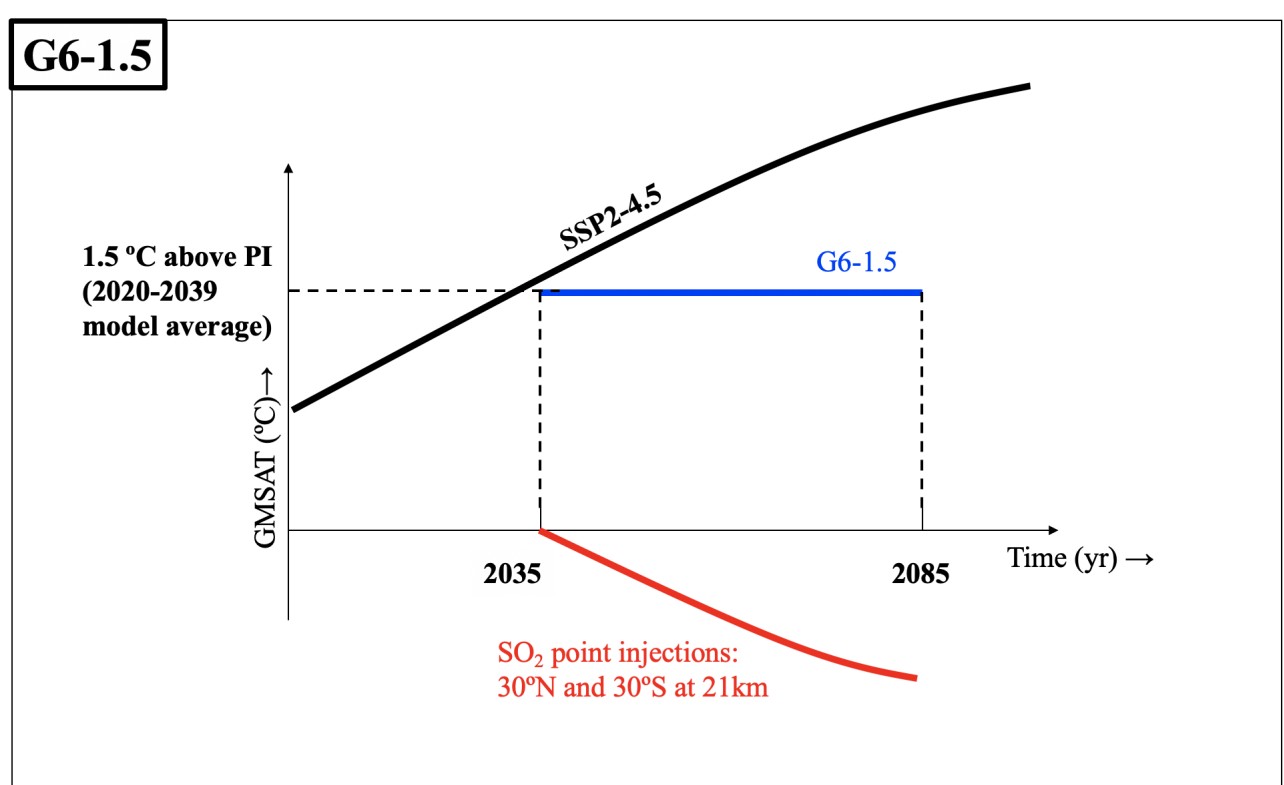

**Figure 3.** A summary of the proposal for the new experiment G6-1.5. The black line represents the global mean surface air temperature (GMSAT) under the underlying emission scenario SSP2-4.5. The blue line represents the temperature under the proposed G6-1.5 experiment. The red line represents the amount of cooling over time. PI=Preindustrial.



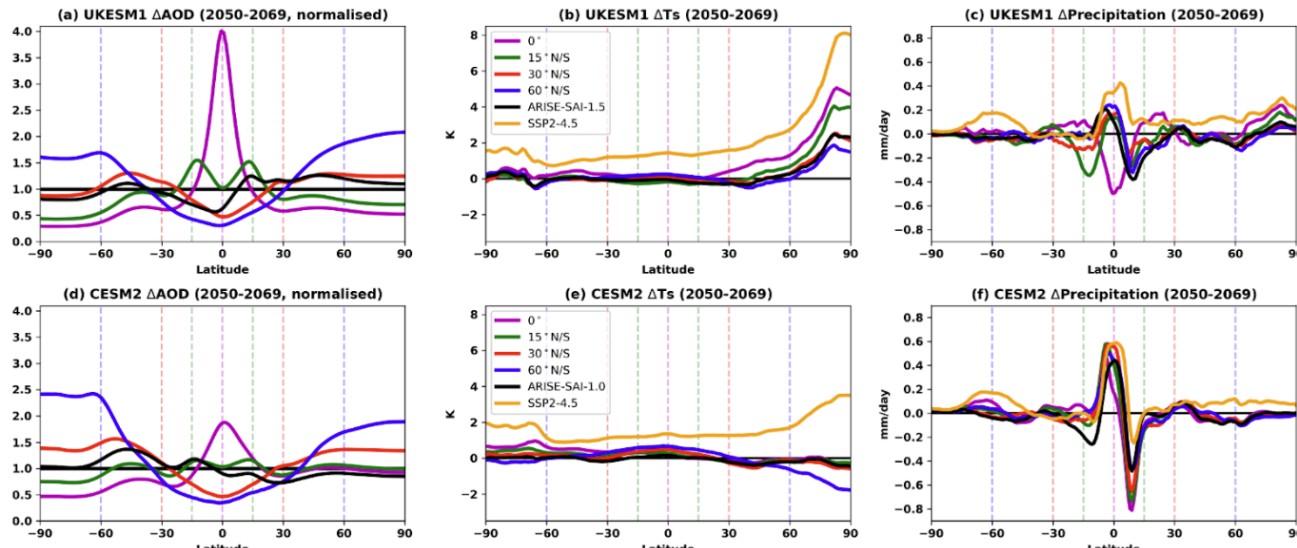

**Figure 4.** A comparison of aerosol optical depth (at 550 nm), surface air temperature change, and precipitation change for two Earth System Models (UKESM1 and CESM2) using different latitudes: injecting everything at the equator (0º), symmetric injection in both hemispheres (15ºN/S, 30ºN/S, and 60ºN/S), or injection at 15ºN, 30ºN, 15ºS, and 30ºS with the objective of maintaining the equator-to-pole and interhemispheric differences in temperature at their reference levels (ARISE-SAI-1.5, Richter et al. 2022, Henry et al. 2023). The target for CESM2 is 0.5ºC below its reference period (2020-39), whereas the target for UKESM1 is 1.5ºC above its preindustrial temperature, which is reached in 2014-2033. Shown are the temperature and precipitation changes with respect to each model's reference period. UKESM1 has 1 ensemble member per experiment whereas CESM2 has 3 ensemble members per experiment.