# Peer review of "G6-1.5K-SAI: a new Geoengineering Model Intercomparison Project (GeoMIP) experiment integrating recent advances in solar radiation modification studies"

_EGUsphere, 2023_

## Author Comment (AC1)

**Reviewers' comments are in black.** Authors responses are in blue.

**This paper describes a proposed new interim GeoMIP experiment for an SAI scenario holding global temperatures to 1.5C above pre-industrial. The paper sets out the trade-offs in the various decisions underlying the experiment, the authors' proposed choices, and their motivations. The paper makes a valuable contribution, which I believe will be much used by the community in running and analysing their proposal for a much-needed addition to the set of GeoMIP experiments. I support its publication given a few minor edits. I make several suggestions below.**

We thank the reviewer for their positive and supportive comments. We address their suggestions below.

**Definition of 1.5°C**

**Consider adding more explanation for the decision to use the 2020-2039 period in each model to define "1.5°C" rather than either of the two options based on temperature anomaly. This choice will, I suspect, be surprising to many, as it means that despite the scenario being called 'G6-1.5K-SAI', the models will be producing simulations of worlds at temperature anomalies anywhere between about 1K and 2K above their own pre-industrial. I suggest that what is currently Section 4.3 needs its own sub-section in your Section 3 clearly laying out the three definitions of 1.5°C, and the pros and cons of each. It's not obvious to me why different start dates between models would make "intermodal comparisons more difficult" (line 238) any more than different temperature anomalies would. So I think some more explanation is needed of the implications of this decision for the eventual intermodal comparisons which are run.**

Figure 2 makes the point that, of all potential metrics, the 2020-2039 average is the one where models present the smallest spread: this is due to them being closer to their historical periods, that models try to match, rather than to PI-control. We note that the IPCC itself uses slightly different definitions of what "above PI" means in different Assessment Reports, and it is usually referred to values such as "1850-1900 average" in observations, but in looking at model future projections the models' own preindustrial values are not typically used and instead models are anchored to observations at some point in recent past (2006-2015 in SR1.5 and 1995-2014 in AR6) and "1.5C" then defined so that the warming up to that period is determined by the observations, not the models. Hence we don't think it would be as surprising to not refer to each model's own preindustrial; nonetheless the choice to anchor off of a future period rather than recent observations differs from the use in IPCC. We have expanded the discussion in the current Section 4.3 to clarify some of these issues: in particular, we point to the similarities with recent discussions in the literature about how to define 1.5ºC operationally (Bett et al., 2023), in our view reinforcing our choice.

**Magnitude of cooling**

**One key decision which is being taken here but not spelled out is the overall magnitude of SAI intervention, as separate from the background emissions scenario. You do mention this, when you describe the potential for lower priority 1°C and 2°C scenarios alongside the 1.5°C one. However, it would be nice to see this discussed as an explicit decision being made, in section 3, since it involves an important trade-off between signal-to-noise ratios and policy relevance.**

We added a point about this in section 3, thank you for the suggestion.

**Figure 1**

**This figure is very useful, but I suggest reworking it somewhat for clarity. Consider including bullet points for separate options under each box, and ensuring consistency that each bullet point represents one possible option for the decision in that box. For example, "studying long term consequences" isn't a possible option for "end date", whereas "end of underlying scenario (2100)" is.**

Thank you for the suggestion, done.

[Figure]

Figure 2

**It would be nice to also include the CMIP6 range on these plots. In the longer term (and looking towards CMIP7), presumably more and more models will start running this and future experiments. So, in the choice of experiment design, we ought to be considering the full CMIP6 spread here, not just the G6 models' spread. This could be added as a box-plot behind the scattered points in panel b, for example.**

**In panel (b) I would have liked to have the values of anomaly relative to PI at 2020-2039 for each model. The reader can just about read this off by comparing the 2nd and 3rd columns of scattered points, but it's a bit tricky to do.**
**Plot (b) y-axis is mislabelled as 'K' rather than 'C' (or '°C')**
**Text is a little small, and the points slightly overlap the title in panel b.**

Thank you, we have modified the figure based on the suggestions. In particular, we have added the CMIP6 average and standard deviation, this was a really good suggestion! Please find the new figure below.

[Figure]

**Figure 4**

**Figure four accompanies the point made in the text at line 265 that injection at 30N/S controlling for T0 is functionally very similar to ARISE controlling for T0/T1/T2, in both CESM2 and UKESM. I wonder if this point (and figure) really belongs in its own paper, with the space to fully make the argument and explore the consequences. It feels a little hidden away here and is only referenced in passing in this paper. In any case, if the authors choose to retain figure 4, perhaps the rows and columns could be flipped (one column per model) to increase the size.**

We have discussed the results a bit more now, but we also mostly clarified that those were results discussed in depth for CESM2 in Zhang et al. (2023) and for at least one strategy in UKESM1 in Henry et al. (2023). We've updated the figure as suggested.

**Other minor points:**

**Consider rephrasing the section titles (e.g. "Reasons behind a new experiment and its timing" and "Required decisions towards a new experiment")**

We haven't really found any better section titles, and are unclear over how the reviewer is suggesting to change them.

**Check consistency in referencing figures. At times you use 'Figure X', but elsewhere it is 'Fig. X'**

Fixed, thank you.

**Line 239, "are evident in Figure 4". Do you mean Figure 2 here?**

Yes, corrected.

References

Betts, R. A., Belcher, S. E., Hermanson, L., Tank, A. K., Lowe, J. A., Jones, C. D., Morice, C. P., Rayner, N. A., Scaife, A. A., and Stott, P. A.: Approaching 1.5 °C: how will we know we've reached this crucial warming mark?, Nature, 624, 33–35, https://doi.org/10.1038/d41586-023-03775-z, 2023.

---

## Author Comment (AC2)

**Visioni et al. proposed a new GeoMIP experiment for the Solar Radiation Management experiment that to be run with CMIP6 Earth system models (ESMs). Their specific design considers the potential application in the future CMIP7 as well. All the details of the new experiment were clearly described, discussed, for example, Target metric, underlying emission scenario, goal, start date, end date, and forcing strategy. Two ESMs were tested with the new experiment to show the impacts of SIC on surface air temperature and mean precipitation. Overall, I think the manuscript is well written, and the new experiment is carefully and clearly described. It is suitable to be published in Geoscientific Model Development. The only major comment I have is that more discussion of the results (e.g., Fig. 3) are needed. Currently, Fig.3 is only discussed in a few lines. In addition, Fig. 2 was not referred in the main text. I suggest more work for the results section for the revision.**

We thank the reviewer for their comments. We apologize for the confusion, but we clarified now in the text that most of the results were already discussed in depth in Zhang et al. (2023) (now accepted and close to final publication) for CESM2, and for some strategies in Henry et al. (2023) in UKESM1. Therefore, we see no necessity here to revisit in depth the results. However, we hope to have clarified this much better now.

**Specific comments**

**Line 147: Please give the full name of PI.**

Done!

**Figure 2 was not referred in the main text.**

Thank you, we fixed this now.

**Line 203 – Line 205: how were these results estimated? Please specify the method of estimating those GMP numbers.**

We added some more specifications right after.